# PRECONDITION LAYER AND ITS USE FOR GANS

## ABSTRACT

One of the major challenges when training generative adversarial nets (GANs) is instability. To address this instability spectral normalization (SN) is remarkably successful. However, SN-GAN still suffers from training instabilities, especially when working with higher-dimensional data. We find that those instabilities are accompanied by large condition numbers of the discriminator weight matrices. To improve training stability we study common linear-algebra practice and employ preconditioning. Specifically, we introduce a preconditioning layer (PC-layer) that performs a low-degree polynomial preconditioning. We use this PC-layer in two ways: 1) fixed preconditioning (FPC) adds a fixed PC-layer to all layers; and 2) adaptive preconditioning (APC) adaptively controls the strength of preconditioning. Empirically, we show that FPC and APC stabilize training of unconditional GANs using classical architectures. On LSUN $256 \times 256$ data, APC improves FID scores by around 5 points over baselines.

## 1 INTRODUCTION

Generative Adversarial Nets (GANs) (Goodfellow et al., 2014) successfully transform samples from one distribution to another. Nevertheless, training GANs is known to be challenging, and its performance is often sensitive to hyper-parameters and datasets. Understanding the training difficulties of GAN is thus an important problem.

Recent studies in neural network theory (Pennington et al., 2017; Xiao et al., 2018; 2020) suggest that the spectrum of the input-output Jacobian or neural tangent kernel (NTK) is an important metric for understanding training performance. While directly manipulating the spectrum of the Jacobian or NTK is not easy, a practical approach is to manipulate the spectrum of weight matrices, such as orthogonal initialization (Xiao et al., 2018). For a special neural net, Hu et al. (2020) showed that orthogonal initialization leads to better convergence result than Gaussian initialization, which provides early theoretical evidence for the importance of manipulating the weight matrix spectrum.

Motivated by these studies, we suspect that an 'adequate' weight matrix spectrum is also important for GAN training. Indeed, one of the most popular techniques for GAN training, spectral normalization (SN) (Miyato et al., 2018), manipulates the spectrum by scaling all singular values by a constant. This ensures the spectral norm is upper bounded. However, we find that for some hyperparameters and for high-resolution datasets, SN-GAN fails to generate good images. In a study we find the condition numbers of weight matrices to become very large and the majority of the singular values are close to 0 during training. See Fig. 1(a) and Fig. 2(a). This can happen as SN does not promote a small condition number.

This finding motivates to reduce the condition number of weights during GAN training. Recall that controlling the condition number is also a central problem in numerical linear algebra, known as preconditioning (see Chen (2005)). We hence seek to develop a "plug-in" preconditioner for weights. This requires the preconditioner to be differentiable. Out of various preconditioners, we find the polynomial preconditioner to be a suitable choice due to the simple differentiation and strong theoretical support from approximation theory. Further, we suggest to adaptively adjust the strength of the preconditioner during training so as to not overly restrict the expressivity. We show the efficacy of preconditioning on CIFAR10 ($32 \times 32$), STL ($48 \times 48$) and LSUN bedroom, tower and living room ($256 \times 256$).

**Summary of contributions.** For a deep linear network studied in (Hu et al., 2020), we prove that if all weight matrices have bounded spectrum, then gradient descent converges to global min-

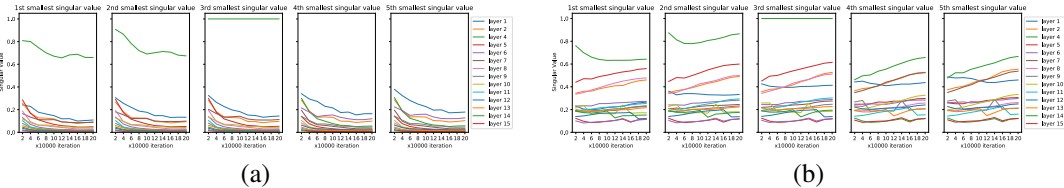

**Figure 1:** Evolution of the 5 smallest singular values of (a) SN-GAN, FID 147.9 and (b) APC-GAN, FID 34.08 when generating STL-10 images with a ResNet trained with $D_{\text{it}} = 1$ for $200k$ iterations. The max singular value is around 1 due to SN, thus not shown here.

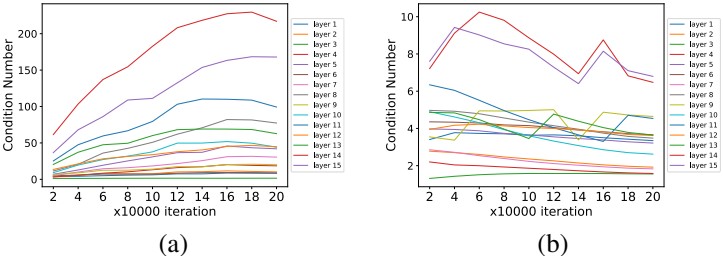

**Figure 2:** Evolution of condition number of (a) SN-GAN, FID 147.9 and (b) APC-GAN, FID 34.08 when generating STL-10 images with a ResNet trained with $D_{\text{it}} = 1$ for $200k$ iterations.

imum at a geometric rate. We then introduce a PC-layer (preconditioning layer) that consists of a low-degree polynomial preconditioner. We further study adaptive preconditioning (APC) which adaptively controls the strength of PC on different layers in different iterations. Applying PC and APC to unconditional GAN training on LSUN data ($256 \times 256$), permits to generate high-quality images when SN-GAN fails. We also show that APC achieves better FID scores on CIFAR10, STL, and LSUN than a recently proposed method of Jiang et al. (2019).

## 1.1 RELATED WORK

Related to the proposed method is work by Jiang et al. (2019), which also controls the spectrum in GAN training. They re-parameterize a weight matrix $W$ via $W = USV^T$, add orthogonal regularization of $U, V$ and certain regularizer of entries of the diagonal matrix $S$. This approach differs from ours in a few aspects. First, Jiang et al. (2019) essentially solves a constrained optimization problem with constraints $U^T U = I, V^T V = I$ using a penalty method (Bertsekas, 1997). In contrast, our approach solves an unconstrained problem since we add one layer into the neural net, similar to batch normalization (BN) (Ioffe & Szegedy, 2015) and SN (Miyato et al., 2018). Second, our PC layer is a direct generalization of SN as it includes SN-layer as a special case. In contrast, the method of Jiang et al. (2019) differs from SN-layer in any case. Our proposed method thus offers a smoother transition for existing users of SN.

In a broader context, a number of approaches have been proposed to stabilize and improve GAN training, such as modifying the loss function (Arjovsky et al., 2017; Arjovsky & Bottou, 2017; Mao et al., 2017; Li et al., 2017b; Deshpande et al., 2018), normalization and regularization (Gulrajani et al., 2017; Miyato et al., 2018), progressive growing techniques (Karras et al., 2018; Huang et al., 2017), changing architecture (Zhang et al., 2019; Karnewar & Wang, 2019), utilizing side information like class labels (Mirza & Osindero, 2014; Odena et al., 2017; Miyato & Koyama, 2018). Using this taxonomy, our approach fits the "normalization and regularization" category (even though our method is not exactly normalization, the essence of "embedded control" is similar). Note that these directions are relatively orthogonal, and our approach can be potentially combined with other techniques such as progressive growing. However, due to limited computational resources, we focus on unconditional GANs using classical architectures, the setting studied by Miyato et al. (2018).

## 1.2 NOTATION AND DEFINITION

We use $\text{eig}(A)$ to denote the multiset (i.e., allow repetition) of all eigenvalues of $A$. If all eigenvalues of $A$ are non-negative real numbers, we say $A$ is a positive semidefinite (PSD) matrix. The singular values of a matrix $A \in \mathbb{R}^{n \times m}$ are the square root of the eigenvalues of $A^T A \in \mathbb{R}^{m \times m}$. Let $\sigma_{\max}(A)$ and $\sigma_{\min}(A)$ denote the maximum and minimum singular values of $A$. Let $\|A\|_2$ denote

the spectral norm of $A$, i.e., the largest singular value. The condition number of a square matrix $A$ is traditionally defined as $\kappa(A) = \|A\|_2\|A^{-1}\|_2 = \frac{\sigma_{\max}(A)}{\sigma_{\min}(A)}$. We extend this definition to a rectangular matrix $A \in \mathbb{R}^{n \times m}$ where $n \geq m$ via $\kappa(A) = \frac{\sigma_{\max}(A)}{\sigma_{\min}(A)}$. Let $\deg(p)$ denote the degree of a polynomial $p$ and let $P_k = \{p \mid \deg(p) \leq k\}$ be the set of polynomials with degree no more than $k$.

## 2 WHY CONTROLLING THE SPECTRUM?

To understand why controlling the spectrum is helpful we leverage recent tools in neural network theory to prove the following result: if weight matrices have small condition numbers, then gradient descent for deep pyramid linear networks converges to the global-min fast. This is inspired by Hu et al. (2020) analyzing a deep linear network to justify orthogonal initialization.

Similar to Hu et al. (2020), we consider a linear network that takes an input $x \in \mathbb{R}^{d_x \times 1}$ and outputs

$$F(\theta; x) = W_L W_{L-1} \ldots W_1 x \in \mathbb{R}^{d_y \times 1}, \tag{1}$$

where $\theta = (W_1, \ldots, W_L)$ represents the collection of all parameters and $W_j$ is a matrix of dimension $d_j \times d_{j-1}$, $j = 1, \ldots, L$. Here we define $d_0 = d_x$ and $d_L = d_y$. Assume there exists $r \in \{1, \ldots, L\}$, such that $d_y = d_L \leq d_{L-1} \leq \cdots \leq d_r$, and $n \leq d_0 \leq d_1 \leq \cdots \leq d_r$. This means the network is a pyramid network, which generalizes the equal-width network of Hu et al. (2020).

Suppose $y = (y_1; \ldots; y_n) \in \mathbb{R}^{n d_y \times 1}$ are the labels, and the predictions are $F(\theta; X) = (F(\theta; x_1); \ldots; F(\theta; x_n)) \in \mathbb{R}^{n d_y \times 1}$. We consider a quadratic loss $\mathcal{L}(\theta) = \frac{1}{2}\|y - F(\theta; X)\|^2$.

Starting from $\theta(0)$, we generate $\theta(k) = (W_1(k), \ldots, W_L(k)), k = 1, 2, \ldots$ via gradient descent:

$$\theta(k+1) = \theta(k) - \eta \nabla \mathcal{L}(\theta(k)).$$

Denote the residual $e(k) = F(\theta(k); X) - y$. For given $\tau_l \geq 1, \mu_l \geq 0, l = 1, \ldots, L$, define

$$R \triangleq \{\theta = (W_1, \ldots, W_L) \mid \tau_l \geq \sigma_{\max}(W_l) \geq \sigma_{\min}(W_l) \geq \mu_l, \ \forall l\}.$$

$$\beta \triangleq L\|X\|_2 \tau_L \ldots \tau_1 \left(\|e(0)\| + \|X\|_F \tau_L \ldots \tau_1\right), \quad \mu \triangleq (\mu_1 \ldots \mu_L)^2 \sigma_{\min}(X)^2.$$

The following result states that if $\theta(k)$ stay within region $R$ (i.e., weight matrices have bounded spectrum) for $k = 0, 1, \ldots, K$, then the loss decreases at a geometric rate until iteration $K$. The rate $(1 - \frac{\beta}{\mu})$ depends on $\frac{(\tau_L \ldots \tau_1)^2}{(\mu_L \ldots \mu_1)^2}$, which is related to the condition numbers of all weights.

**Theorem 1** *Suppose $\eta = \frac{1}{\beta}$. Assume $\theta(k) \in R$, $k = 0, 1, \ldots, K$. Then we have*

$$\|e(k+1)\|^2 \leq (1 - \frac{\mu}{\beta})\|e(k)\|^2, \quad k = 0, 1, \ldots, K. \tag{2}$$

See Appendix D.3.1 for the proof and detailed discussions.

For proper initial point $\theta(0)$ where $W_l(0)$'s are full-rank, we can always pick $\tau_l, \sigma_l$ so that $\theta(0) \in R$. The trajectory $\{\theta(k)\}$ either stays in $R$ forever (in which case $K = \infty$), or leaves $R$ at some finite iteration $K$. In the former case, the loss converges to zero at a geometrical rate; in the latter case, the loss decreases to below $(1 - \mu/\beta)^K\|e(0)\|^2$. However, our theorem does not specify how large $K$ is for a given situation. Previous works on convergence (e.g., Hu et al., 2020; Du et al., 2018; Allen-Zhu et al., 2019; Zou et al., 2018) bound the movement of the weights with extra assumptions, so that the trajectory stays in a certain nice regime (related to $R$). We do not attempt to prove the trajectory stays in $R$. Instead, we use this as a motivation for algorithm design: if we can improve the condition numbers of weights during training, then the trajectory may stay in $R$ for a longer time, and thus lead to smaller loss values. Next, we present the preconditioning layer as such a method.

## 3 PRECONDITIONING LAYER

In the following, we first introduce classical polynomial preconditioners in Sec. 3.1. We then present the preconditioning layer for deep nets in Sec. 3.2. We explain how to compute a preconditioning polynomial afterwards in Sec. 3.3, and finally present an adaptive preconditioning in Sec. 3.4.

### 3.1 PRELIMINARY: POLYNOMIAL PRECONDITIONER

Preconditioning considers the following classical question: for a symmetric matrix $Q$, how to find an operator $g$ such that $\kappa(g(Q))$ is small? Due to the importance of this question and the wide applicability there is a huge literature on preconditioning. See, e.g., Chen (2005) for an overview, and Appendix B for a short introduction. In this work, we focus on polynomial preconditioners (Johnson et al., 1983). The goal is to find a polynomial $\hat{p}$ such that $\hat{p}(Q)Q$ has a small condition number. The matrix $\hat{p}(Q)$ is often called *preconditioner*, and $\hat{g}(Q) \triangleq \hat{p}(Q)Q$ is the *precondtioned* matrix. We call $g$ the *preconditioning polynomial*. Polynomial preconditioning has a special merit: the difficult problem of manipulating eigenvalues can be transformed to manipulating a 1-d function, based on the following fact (proof in Appendix E.2.1).

**Claim 3.1** *Suppose $\hat{g}$ is any polynomial, and $Q \in \mathbb{R}^{m \times m}$ is a real symmetric matrix with eigenvalues $\lambda_1 \leq \cdots \leq \lambda_m$. Then the eigenvalues of the matrix $\hat{g}(Q)$ are $\hat{g}(\lambda_i), i = 1, \ldots, m$. As a corollary, if $\hat{g}([\lambda_1, \lambda_m]) \subseteq [1 - \epsilon, 1]$, then $eig(\hat{g}(Q)) \subseteq [1 - \epsilon, 1]$.*

To find a matrix $\hat{g}(Q) = \hat{p}(Q)Q$ that is well-conditioned, we need to find a polynomial $\hat{p}$ such that $\hat{g}(x) = \hat{p}(x)x$ maps $[\lambda_1, \lambda_m]$ to $[1 - \epsilon, 1]$. This can be formulated as a function approximation problem: find a polynomial $\hat{g}(x)$ of the form $x\hat{p}(x)$ that approximates a function $\hat{f}(\lambda)$ in $\lambda \in [\lambda_1, \lambda_m]$. Under some criterion, the optimal polynomial is a variant of the Chebychev polynomial, and the solutions to more general criteria are also well understood. See Appendix B.1 for more.

A scaling trick is commonly used in practice. It reduces the problem of preconditioning $Q$ to the problem of preconditioning a scaled matrix $Q_{\text{sca}} = Q/\lambda_m$. Scaling employs two steps: first, we find a polynomial $g$ that approximates $f(x) = 1$ in $x \in [\lambda_1/\lambda_m, 1]$; second, set $\hat{g}(\lambda) = g(\lambda/\lambda_m)$ and use $\hat{g}(Q) = g(Q/\lambda_m) = g(Q_{\text{sca}})$ as the final preconditioned matrix. It is easy to verify $\hat{g}$ approximates $\hat{f}(\lambda) = 1$ in $[\lambda_1, \lambda_m]$. Thus this approach is essentially identical to solving the approximation problem in $[\lambda_1, \lambda_m]$. Johnson et al. (1983) use this trick mainly to simplify notation since they can assume $\lambda_m = 1$ without loss of generality. We will use this scaling trick for a different purpose (see Section 3.3).

### 3.2 PRECONDITIONING LAYER IN DEEP NETS

Suppose $D(W_1, \ldots, W_L)$ is a deep net parameterized by weights $W_1, \ldots, W_L$ for layers $l \in \{1, \ldots, L\}$. To control the spectrum of a weight $W_l$, we want to embed a preconditioner $\hat{g}$ into the neural net. Among various preconditioners, polynomial ones are appealing since their gradient is simple and permits natural integration with backpropagation. For this we present a preconditioning layer (PC-layer) as follows: a PC-layer $\hat{g}(W) = g(\text{SN}(W))$ is the concatenation of a preconditioning polynomial $g$ and an SN operation of (Miyato et al., 2018) (see Appendix app sub: details of FPC and APC for details of $\text{SN}(W)$). The SN operator is used as a scaling operator (reason explained later). We describe an efficient implementation of PC-layer in Appendix C.3.

In our case, we use $A = \text{SN}(W)$ to indicate the scaled matrix. Prior work on polynomial preconditioners (Johnson et al., 1983; Chen, 2005) often study square matrices. To handle rectangular matrices, some modifications are needed.

A naïve solution is to apply a preconditioner to the symmetrized matrix $A^T A$, leading to a matrix $g(A) = p(A^T A)A^T A$. This solution works for linear models (see Appendix B.2 for details), but it is not appropriate for deep nets since the shape of $p(A^T A)A^T A \in \mathbb{R}^{m \times m}$ differs from $A$. To maintain the shape of size $n \times m$, we propose to transform $A$ to $g(A) = p(AA^T)A \in \mathbb{R}^{n \times m}$. This transformation works for general parameterized models including linear models and neural nets. For a detailed comparison of these two approaches, see Appendix B.2. The following claim relates the spectrum of $A$ and $p(AA^T)A$; see the proof in Appendix E.2.2.

**Claim 3.2** *Suppose $A \in \mathbb{R}^{n \times m}$ has singular values $\sigma_1 \leq \cdots \leq \sigma_m$. Suppose $g(x) = p(x^2)x$ where $p$ is a polynomial. Then the singular values of $g(A) = p(AA^T)A$ are $|g(\sigma_1)|, \ldots, |g(\sigma_m)|$.*

To find a polynomial $p$ such that $g(A) = p(AA^T)A$ is well-conditioned, we need to find a polynomial $p$ such that $g(x) = p(x^2)x$ maps $[\sigma_1, \sigma_m]$ to $[1 - \epsilon, 1]$ for some $\epsilon$. This can be formulated as a function approximation problem: find a polynomial $g(x)$ in $G_k$ that approximates a function

$f(\sigma) = 1$ in $\sigma \in [\sigma_1, \sigma_m]$ where $G_k = \{g(x) = xp(x^2) \mid p \in P_k\}$. We will describe the algorithm for finding the preconditioning polynomial $g$ in Sec. 3.3.

In principle, the PC-layer can be added to any deep net including supervised learning and GANs. Here, we focus on GANs for the following reason. Current algorithms for supervised learning already work quite well, diminishing the effect of preconditioning. In contrast, for GANs, there is a lot of room to improve training. Following SN-GAN which applies SN to the discriminator of GANs, in the experiments we apply PC to the discriminator.

### 3.3 FINDING PRECONDITIONING POLYNOMIALS

In this subsection, we discuss how to generate preconditioning polynomials. This generation is done off-line and independent of training. We will present the optimization formulation and discuss the choice of a few hyperparameters such as the desirable range and the target function $f$.

**Optimization formulation**. Suppose we are given a range $[\gamma_L, \gamma_U]$, a target function $f$, and an integer $k$; the specific choices are discussed later. Suppose we want to solve the following approximation problem: find the best polynomial of the form $g(x) = x(a_0 + a_1 x^2 + \cdots + a_k x^{2k})$ that approximates $f(x)$ in domain $[\gamma_L, \gamma_U]$, i.e., solve

$$\min_{g \in G_k} d_{[\gamma_L, \gamma_U]}(g(x), f(x)), \tag{3}$$

where $G_k = \{g(x) = xp(x^2) \mid p \in P_k\}$, $d_{[\gamma_L, \gamma_U]}$ is a distance metric on the function space $C[\gamma_L, \gamma_U]$, such as the $\ell_\infty$ distance $d_{[\gamma_L, \gamma_U]}(f, g) = \max_{t \in [\gamma_L, \gamma_U]} |f(t) - g(t)|$. We consider a weighted least-square problem suggested by Johnson et al. (1983):

$$\min_{g \in G_k} \int_{\gamma_L}^{\gamma_U} |g(x) - f(x)|^2 w(x) dx, \tag{4}$$

where $w(x) = x^\alpha$ is a weight function used in (Johnson et al., 1983). We discretize the objective and solve the finite-sample version of Eq. (4) as follows:

$$\min_{c=(c_0, c_1, \ldots, c_k) \in \mathbb{R}^{k+1}} \sum_{i=1}^{n} \left| x_i \sum_{t=0}^{k} c_t x_i^{2t} - f(x_i) \right|^2 w(x_i), \tag{5}$$

where $x_i \in [\gamma_L, \gamma_U], \forall i$ (e.g., drawn from uniform distribution on $[\gamma_L, \gamma_U]$). This is a weighted least squares problem (thus convex) that can be easily solved by a standard solver.

**Choice of desirable range** $[\gamma_L, \gamma_U]$. The range $[\gamma_L, \gamma_U]$ within which we want to approximate the target function is often chosen to be the convex hull of the singular values of the matrix to be preconditioned. For the original matrix $W$, the desirable range $[\gamma_L, \gamma_U] = [\sigma_{\min}(W), \sigma_{\max}(W)]$. However, this range varies across different layers and different iterations. For this reason we scale each $W$ by $1/\|W\|_2$ to obtain $A$ so that its singular values lie in a fixed range $[0, 1]$. Note that a more precise range is $[\sigma_{\min}(A)/\sigma_{\max}(A), 1]$, but we can relax it to $[0, 1]$. We follow Miyato et al. (2018) to use one power iteration to estimate the spectral norm $\tilde{W} \approx \|W\|_2$, and denote $\text{SN}(W) = W/\tilde{W}$. Since $\tilde{W}$ is not exactly $\|W\|_2$, the range of singular values of $A = \text{SN}(W)$ may not be exactly in $[0, 1]$. We have checked the empirical estimation and found that the estimated spectral norm during the training of SN-GAN is often less than 1.1 times the true spectral norm (see Fig. 5 in Appendix E.1), thus we pick $[\gamma_L, \gamma_U] = [0, 1.1]$ in our implementation.

**Choice of target function** $f(\lambda)$. Previously, we discuss the ideal situation that $[\gamma_L, \gamma_U] = [\lambda_1, \lambda_m]$, thus the target function is 1. In the previous paragraph, we have relaxed the desirable range to $[0, \gamma_U]$, then we cannot set $f(x) = 1$ in $[0, \gamma_U]$, because any polynomial $g(\lambda) \in G_k$ must satisfy $g(0) = 0$, causing large approximation error at $\lambda = 0$. We shall set $f(0) = 0$. A candidate target function is $\text{PL}_b(x)$, where $\text{PL}_b(x) = \begin{cases} x/b, & x < b \\ 1, & x \geq b \end{cases}$ is defined as a piece-wise linear function with cutoff point $b$. If the cutoff point $b < \lambda_{\min}(A)$, then $\text{PL}_b(\lambda)$ maps all singular values of $A$ to 1.

While setting all singular values to 1 is ideal for fast training, this may reduce the expressiveness for deep nets. More specifically, the set of functions $\{D(W_1, \ldots, W_L) \mid \text{eig}(W_l^T W_l) \subseteq \{1\}\}$ is smaller than $\{D(W_1, \ldots, W_L) \mid \text{eig}(W_l^T W_l) \subseteq [\lambda_0, 1]\}$, thus forcing all singular values to be 1

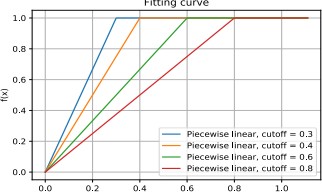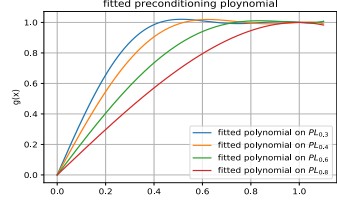

**Figure 3:** Left: Different piecewise linear functions; Right: the corresponding fitted preconditioning polynomials. The smaller the cutoff, the more aggressive the preconditioner. For instance, when the cutoff is $0.3$, the preconditioner pushes all singular values above $0.3$ to $1$. We show the fitting polynomials of degree $3, 5, 7, 9$ for $PL_{0.8}, PL_{0.6}, PL_{0.4}, PL_{0.3}$ respectively. More details are in Sec. C.2.

may hurt the representation power. Therefore, we do not want the target function to have value $1$ in $[\lambda_{\min}(A), \gamma_U]$. In practice, the value of $\lambda_{\min}(A)$ varies for different problems, therefore we permit a flexible target function $f$, to be chosen by a user.

In our implementation, we restrict target functions to a family of piece-wise linear functions. We use $PL_b(x)$ with a relatively large cutoff point $b$, such as $0.8$ and $0.3$. We plot our candidate target functions $PL_{0.3}$, $PL_{0.4}$, $PL_{0.6}$ and $PL_{0.8}$ in Figure 3. As the cutoff point $b$ changes from 1 to 0, the function $PL_b$ becomes more aggressive as it pushes more singular values to $1$. As a result, the optimization will likely become easier, while the representation power becomes weaker. The exact choice of the target function is likely problem-dependent, and we discuss two strategies to select them in Section 3.4.

**Search space of preconditioning polynomial**. As mentioned earlier, the default search space is $G_k = \{g(x) = xp(x^2) \mid p \in P_k\}$ for a pre-fixed $k$. The degree of $g(\lambda)$ is an important hyperparameter. On the one hand, the higher the degree, the better the polynomial can fit the target function $f$. On the other hand, higher degree leads to more computation. In our implementation, we consider $k = 1, 2, 3, 4$, i.e., polynomials of degree $3, 5, 7$ and $9$. The extra time is relatively small; see Section C.4 for details.

### 3.4 FIXED PRECONDITIONING AND ADAPTIVE PRECONDITIONING

The preconditioning polynomial can be determined by the target function and the degree $k$. Which polynomial shall we use during training?

**Candidate preconditioners**. At first sight, there are two hyper-parameters $b$ and $k$. Nevertheless, if $b$ is small (steep slope), then it is hard to approximate $PL_b$ by low-order polynomials. For each degree $k \in \{3, 5, 7, 9\}$, there is a certain $b_k$ such that $b < b_k$ leads to large approximation error. We find that $b_3 \approx 0.8, b_5 \approx 0.6, b_7 \approx 0.4, b_9 \approx 0.3$. After fitting $PL_{0.3}$, $PL_{0.4}$, $PL_{0.6}$ and $PL_{0.8}$, the resulting polynomials are shown in Figure 3.

A natural approach is to add the PC-layer to all layers of the neural net, resulting in a preconditioned net $D_{PC}(\theta) = D(g(SN(W_1)), \ldots, g(SN(W_L)))$. We call this method fixed preconditioning (FPC). Just like other hyperparameters, in practice, we can try various preconditioners and pick the one with the best performance. Not surprisingly, the best preconditioner varies for different datasets.

**Adaptive preconditioning (APC)**. Motivated by adaptive learning rate schemes like Adam (Kingma & Ba, 2014) and LARS (You et al., 2017), we propose an adaptive preconditioning scheme. In APC, we apply the preconditioner in a epoch-adaptive and layer-adaptive manner: at each epoch and for each layer the algorithm will automatically pick a proper preconditioner based on the current condition number.

The standard condition number $\kappa(A) = \frac{\sigma_{max}(A)}{\sigma_{\min}(A)}$ is not necessarily a good indicator for the optimization performance. In APC, we use a modified condition number $\tilde{\kappa}(A) = \frac{\sigma_{max}(A)}{(\sum_{i=1}^{m_0} \sigma_i(A))/m_0}$. where $A$ has $m$ columns and $m_0 = \lceil \frac{m}{10} \rceil$. We prepare $r$ preconditioning polynomials $g_1, \ldots, g_r$ with different strength (e.g., the four polynomials $g_1, g_2, g_3, g_4$ shown in Figure 3). We set a number of ranges $[0, \tau_1], [\tau_1, \tau_2], \ldots, [\tau_r, \infty]$ and let $\tau_0 = 0, \tau_{r+1} = \infty$. If the modified condition number of $A$ falls into the range $[\tau_i, \tau_{i+1}]$ for $i \in \{0, 1, \ldots, r\}$, we will use $g_i$ in the PC-layer. In our im-

plementation, we set $r = 4$. To save computation, we only compute the modified condition number and update PC strength at a fixed interval (e.g., every 1000 iterations). The summary of APC is presented in Table 3 in Appendix C.2.

**Computation time.** We use a few implementation tricks; see Appendix C.3. In our implementation of FPC with a degree $3, 5, 7$ or $9$ polynomial, the actual added time is around $20 - 30\%$ (Fig. 4 (a)) of the original training time of SN-GAN. Fig. 4 (b) shows that the extra time of APC over SN is often less than $10\%$. See Appendix C.4 for more on the computation time.

## 4 EXPERIMENTAL RESULTS

We will demonstrate the following two findings. First, SN-GAN still suffers from training instabilities, and the failure case is accompanied by large condition numbers. Second, PC layers can reduce the condition number, and improve the final performance, especially for high resolution data (LSUN $256 \times 256$).

We conduct a set of experiments for unconditional image generation on CIFAR-10 ($32 \times 32$), STL-10 ($48 \times 48$), LSUN-bedroom ($128 \times 128$ and $256 \times 256$), LSUN-tower ($256 \times 256$) and LSUN-living-room ($256 \times 256$). We also compare the condition numbers of the discriminator layers for different normalization methods to demonstrate the connection between the condition number and the performance. The following methods are used in our experiments: standard SN; SVD with D-Optimal Reg. (Jiang et al., 2019); FPC with degree 3 or 7 preconditioners; APC. Following Miyato et al. (2018), we use the log loss GAN on the CNN structure and the hinge loss GAN on the ResNet structure.

**CIFAR and STL: Training failure of (1,1)-update**. Tuning a GAN is notoriously difficult and sensitivity to hyper-parameters. Even for low-resolution images, without prior knowledge of good hyper-parameters such as $D_{\text{it}}, G_{\text{it}}$, training a GAN is often not trivial. On CIFAR10, SN-GAN uses $D_{\text{it}} = 5, G_{\text{it}} = 1$ for ResNet; for simplicity, we call it a $(5, 1)$-update. However, using a $(1, 1)$-update, i.e., changing $D_{\text{it}} = 5$ to $D_{\text{it}} = 1$ while keeping $G_{\text{it}} = 1$, will lead to an SN-GAN training failure: a dramatic decrease of final performance and an FID score above 77. SN-GAN with $(1, 1)$-update also fails on STL data, yielding an FID score above 147. We are interested in stabilizing the $(1, 1)$-update for two reasons: first, trainability for both $(1, 1)$-update and $(5, 1)$-update means improved training stability; second, the $(1, 1)$-update requires only about $1/3$ of the time compared to the $(5, 1)$-update. Therefore, in the first experiment, we explore GAN-training with $(1, 1)$-update on CIFAR-10 and STL-10.

**Failure mode: large condition numbers**. Understanding the failure mode of training is often very useful for designing algorithms (e.g., Glorot & Bengio, 2010). We suspect that a large condition number is a failure mode for GAN training. As Table 1 shows, the high FID scores (bad case) of SN-GAN are accompanied by large condition numbers.

**PC reduces condition numbers and rectifies failures**. Table 1 shows that FPC and APC can both greatly improve the training performance: they reduce FID from 77 to less than 20 for CIFAR-10, and reduce FID from 147 to less than 34 for STL in $200k$ iterations. The evolution of the 5 smallest singular values of the adaptive preconditioned matrices and the condition numbers are showed in Fig. 1(b) and Fig. 2(b) for STL-10 training on ResNet with $D_{\text{it}} = 1$. This shows that PC-GAN successfully improves the spectrum of weight matrices in this setting.

**Experiments on "good" case of SN-GAN**. We report the results for the $(5, 1)$-update on CIFAR-10 and STL-10 with ResNet in the Appendix. For those FPC and APC achieve similar or slightly better FID scores. We also report IS scores there. We also list the results of PC and multiple baselines on the CNN structure in the Appendix.

**High resolution images LSUN.** Using high-resolution data is more challenging. We present numerical results on LSUN bedroom ($128 \times 128$, and $256 \times 256$) , LSUN tower ($256 \times 256$) and LSUN living room ($256 \times 256$) data in Table 2. The training time for one instance is 30 hours on a single RTX 2080 Ti (200k iterations).

Note, SN-GAN is unstable and results in FID $> 80$ for LSUN-bedroom $256 \times 256$. The SVD method, our FPC and APC generate reasonable FID scores on all three data sets. Importantly, our FPC is comparable or better than SVD, and our APC consistently outperforms the SVD method by

| Setting | Method | FID score | $\max_{l=1}^{L} \tilde{\kappa}(\bar{W}_l)$ | 2nd $\max_{l=1}^{L} \tilde{\kappa}(\bar{W}_l)$ | Avg $\tilde{\kappa}(\bar{W}_l)$ |
|---|---|---|---|---|---|
| CIFAR-10 | SN-GAN | 77.82 | 113.55 | 19.46 | 17.77 |
| ResNet | FPC, deg-3 | 20.09 | 22.65 | 1.88 | 3.16 |
| $D_{\text{it}} = 1$ | FPC, deg-7 | 19.31 | 9.88 | 1.32 | 1.85 |
| | APC | 19.53 | 12.06 | 2.81 | 3.19 |
| STL-10 | SN-GAN | 147.90 | 217.06 | 167.97 | 53.37 |
| ResNet | FPC, deg-3 | 33.99 | 23.75 | 22.35 | 4.36 |
| $D_{\text{it}} = 1$ | FPC, deg-7 | 34.28 | 2.55 | 2.72 | 1.53 |
| | APC | 34.08 | 6.79 | 6.47 | 3.90 |

**Table 1:** Comparison of SN-GAN and PC-GAN, using ResNet with $D_{\text{it}} = 1$. Here $\bar{W}_l$ is the preconditioned weighted matrix (i.e., after applying preconditioning). "2nd $\max_{l=1}^{L} \tilde{\kappa}(\bar{W}_l)$" indicates the second largest condition number of all layers. "Avg $\tilde{\kappa}(\bar{W}_l)$" indicates the average of all layer condition numbers.

| Setting | Method | FID score | $\max_{l=1}^{L} \tilde{\kappa}(\bar{W}_l)$ | 2nd $\max_{l=1}^{L} \tilde{\kappa}(\bar{W}_l)$ | Avg $\tilde{\kappa}(\bar{W}_l)$ |
|---|---|---|---|---|---|
| | SN | 53.48 | 5.00 | 2.20 | 2.14 |
| LSUN | GaTech | 50.92 | 1.75 | 1.55 | 1.46 |
| Bedroom 128 | PC, deg-3 | 51.05 | 1.23 | 1.15 | 1.09 |
| | APC | 45.32 | 2.75 | 2.40 | 1.82 |
| | SN | 81.09 | 26.19 | 6.02 | 6.45 |
| LSUN | GaTech | 36.89 | 2.76 | 2.18 | 1.82 |
| Bedroom 256 | PC, deg-3 | 35.61 | 3.04 | 2.03 | 1.64 |
| | APC | 31.17 | 4.35 | 2.80 | 2.18 |
| | SN | 193.75 | 66.87 | 9.86 | 13.54 |
| LSUN | GaTech | 32.79 | 2.00 | 1.95 | 1.60 |
| Living Room 256 | PC, deg-3 | 28.20 | 2.95 | 2.91 | 1.64 |
| | APC | 28.29 | 3.99 | 3.63 | 2.23 |
| | SN | 33.10 | 12.75 | 3.48 | 3.50 |
| LSUN | GaTech | 35.05 | 2.83 | 2.36 | 1.85 |
| Tower 256 | PC, deg-3 | 30.81 | 2.43 | 1.45 | 1.33 |
| | APC | 29.58 | 3.38 | 2.53 | 1.91 |

**Table 2:** Results on LSUN data.

4-6 FID scores in most cases. Also note, the condition numbers of the failure case of SN-GAN are much higher than the two normal cases of SN-GAN. In all cases, FPC and APC achieve significantly lower condition numbers than SN-GAN. APC achieves higher condition numbers than FPC, and also better FID scores. We suspect that FPC over-controls the condition numbers which leads to lower representation power. In contrast, APC strikes a better balance between representation and optimization than FPC. The generated image samples are presented in Appendix F.5.

## 5 CONCLUSION

We prove that for a deep pyramid linear networks, if all weight matrices have bounded singular values throughout training, then the algorithm converges to a global minimal value at a geometric rate. This result indicates that small weight matrix condition numbers are helpful for training. We propose a preconditioning (PC) layer to improve weight matrix condition numbers during training, by leveraging tools from polynomial preconditioning literature. It is differentiable, and thus can be plugged into any neural net. We propose two methods to utilize the PC-layer: in FPC (fixed preconditioning), we add a fixed PC-layer to all layers; in APC (adaptive preconditioning), we add PC-layers with different preconditioning power depending on the condition number. Empirically, we show that applying FPC and APC to GAN training, we can generate good images in a few cases that SN-GAN perform badly, such as LSUN-bedroom $256 \times 256$ image generation.

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
