# OpenReview forum: "Precondition Layer and Its Use for GANs"
_ICLR.cc/2021/Conference — Reject_

### Official Review · AnonReviewer1 · 2020-10-27
**Good paper with clear motivation, nice approach, and good results**

**Rating:** 7
**Confidence:** 4

**Review:**

Update:
Thank the authors for the detailed feedback. I decide to keep the score.
---
---
The paper shows that the failure mode of spectral normalization (SN) is often accompanied by large condition numbers in the discriminator layers. Motivated from this observation, the paper proposes to control the condition numbers of the discriminator layers, by adding preconditioning to the weights. The results show that the proposed approach makes the training more stable and achieves better sample quality on several datasets.

Overall, I enjoy reading this paper. The motivation is clearly explained, and the approach is simple yet effective. I would recommend an accept. I do find that the writing needs to be improved (e.g. there are many typos in both the main paper and the appendix), and more experiments can be explored. However, I think these are relatively minor issues that do not diminish the overall quality of the paper.

The minor issues and suggestions are listed below.

Missing details:
* Section 3.4 mentioned that "the best preconditioner varies for different datasets". It is fine, but you should show the FPC results with all different degrees, so that it is clearer how sensitive FPC is to the degree. I only see the results with degree 3 and 7 in the main text and the appendix.
* APC needs to use the singular values of the weights. How do you compute the singular values, by exact computation or estimation? If you do the exact computation, then I have a question about the scalability of the approach. For convolutional layers (which are the dominant components in the network architectures you experimented on), the computation is cheap because the kernels are small. But it might computationally expensive for fully connected layers.
* How do you compute the actual true spectral norm in Appendix E.1? By exact computation or estimation?

Suggested experiments/discussions/related work:
* From the appendix, I understand that for convolution layers you conduct the preconditioning on the reshaped kernels. However, the condition numbers/singular values of this reshaped kernel are NOT the same as those of the convolution layer (see the discussions in [1]). I wonder how your theorems and results would change considering this difference (e.g. how the theorems would change if you are controlling the condition numbers of the reshaped kernels instead of the layers, and how the results would be if you strictly controlling the condition number of convolutional layers)?
* The spectral normalization paper [2] shows that spectral normalization is robust across different learning rates and betas in the Adam optimizer. You only tried one learning rate and one beta for each experiment. How sensitive are FPC and APC to the learning rates and betas?
* I realize that this paper is very relevant to [1] (which is online recently so I understand that you are not required to know it). That paper also found out the instability issue in different variants of spectral normalization, and proposed an improved version (which are different from yours but could be relevant). Their results on CIFAR-10 and STL-10 seem to be better than FPC and APC you proposed in Table 4. You might want to add the discussions and/or experimental comparisons to it.
* Another missing related work is [3], which discussed the importance of condition numbers in the generators of GANs. They proposed Jacobian Clamping for controlling the condition numbers by regularization. Although they focused on generators, nothing stops it from applying Jacobian Clamping on discriminators. You might want to add the discussions and/or experimental comparisons to this approach.

Writing:
* It is weird to discuss the scaling trick twice in Section 3.1 and in the "Choice of desirable range" paragraph of Section 3.3.
* Before the "Choice of target function" paragraph of Section 3.3, the scaling trick is already discussed, and the previous paragraph ends with [\gamma_L,\gamma_U]=[0,1.1]. It is weird to go back to [\gamma_L,\gamma_U]=[\lambda_1,\lambda_m] again in this paragraph. It is better to stick to the scaled version of singular values in this paragraph.
* It is better the discuss briefly how SVD works in the main text as you are comparing with it.

Typos:
* 6th line in "Choice of desirable range" paragraph in Section 3.3: \sigma_{min}(A) -> \sigma_{min}(A)/\sigma_{max}(A)
* The end of "Search space of preconditioning polynomial" paragraph in Section 3.3: wrong latex code for the section reference?
* Last line before Section 4: compputation -> computation
* Paragraph "Failure mode: large condition numbers." in Section 4: missing a period at the end
* Header of table 1: there should be a bar on top of each W_l
* Caption of table 1: there should be a bar on top of the W_l
* Header of table 2: there should be a bar on top of the W_l
* The last two paragraphs on page 18 & the first paragraph on page 19 (appendix): Table -> Algorithm
* The first paragraph in Appendix F.1: deg-9 -> deg-7
* The last paragraph on page 28 (appendix): mojority -> majority

[1] Lin, Zinan, Vyas Sekar, and Giulia Fanti. "Why Spectral Normalization Stabilizes GANs: Analysis and Improvements." arXiv e-prints (2020).

[2] Miyato, Takeru, et al. "Spectral normalization for generative adversarial networks." arXiv preprint arXiv:1802.05957 (2018).

[3] Odena, Augustus, et al. "Is generator conditioning causally related to gan performance?." arXiv preprint arXiv:1802.08768 (2018).

---

> ### Author Response · Authors · 2020-11-25
> **Response to Review#1**
>
> We thank the reviewer for the helpful feedback.
>
> **Comment:** You should show the FPC results with all different degrees so that it is clearer how sensitive FPC is to the degree.
> **Response:** Thanks. We add the FPC results with degrees 3, 5, 7 and 9 on CIFAR-10 and LSUN-bedroom (256 x 256) to Appendix Sec. F.4 of the revised paper. We observe that FPC-GAN is robust to the degree overall. But the optimal degree requires tuning, motivating the use of APC-GAN.
>
> ---
>
> **Comment:** How do you compute the singular values, by exact computation or estimation?
> **Response:** We compute the singular values exactly (but once every 1000 iterations). The discriminators in the paper only use an FC layer in the end with the output dimension=1. This means that the layer only has 1 singular value with value=1 after the Spectral Normalization. Thus the preconditioner has no effect on that FC layer. For GAN training, exact computation is feasible: the extra running time of APC-GAN compared to SN-GAN is no more than 10% as shown in Fig. 4b.
> We agree that computing all singular values for the fully connected layer is costly when the output has a large dimension, in which case we may need to approximate, e.g., compute a subset of the singular values or use a subset of samples for computation.
>
> ---
>
> **Comment:** How do you compute the actual true spectral norm in Appendix E.1? By exact computation or estimation?
> **Response:** We compute the actual spectral norm using torch.svd(reshaped weight matrix). We believe the embedded function torch.svd provides a rather precise computation since the dimension of the reshaped weight matrix is not very large (in the order of hundreds).
>
> ---
>
> **Comment:** how your theorems and results would change when it comes to the singular values of the convolutional layers?
> **Response:** Thank you for the insightful comment! The theorem is derived for fully connected neural nets, following most other works in the theory literature. There are three differences for CNNs: first, the weight at each layer is a tensor; second, the weight tensor is sparse; third, some parameters are shared. These reasons make the analysis much more complicated, which is why there are few, if any, optimization results in the literature for linear CNNs.
> As for our proof: our proof does not assume a “dense matrix”, so sparsity is not an issue. What remains unclear are the “tensor” and “weight sharing” parts (which might be possible to resolve with a careful treatment). We believe deriving a similar result for CNNs is a very interesting future work.
>
> ---
>
> **Comment:** The spectral normalization paper [2] shows that spectral normalization is robust across different learning rates and betas in Adam optimizer. You only tried one learning rate and one beta for each experiment. How sensitive are FPC and APC to the learning rates and betas?
> **Response:** Thanks for the comment. We conducted the robustness experiments on 8 settings (4 for learning rate choices, 4 for beta choices) in the revised version. The results are listed in Appendix F.3. Overall, we show the APC-GAN is equally robust to or more robust than SN-GAN.
>
> ---
>
> **Comment:** discussions and/or experimental comparisons to related work [1].
> **Response:** Thank you for pointing out this very interesting work. We add it in the related work in the Appendix (page 14 blue part).
>
> ---
>
> **Comment:** discussions and/or experimental comparisons to related work [3].
> **Response:** Thanks for the reference. We add it in the Appendix (page 14 blue part).
> For an empirical comparison: we calculate the condition number of the input-output Jacobian studied by [3] in all the experiments on CIFAR-10 and STL-10 with SN-GAN, PC and APC generators. Interestingly, the condition number constraint on the generator proposed by [3] is satisfied by all these models (i.e., the loss proposed by [3] is 0 in all these models). Thus combining the algorithm of [3] and SN-GAN would be equivalent to SN-GAN. This is understandable since [3] appeared around the same time as SN-GAN, so they did not know their method would be redundant in an SN-GAN.
>
> ---
>
> **Comment:** Writing & typos suggestions.
> **Response:** Thanks for pointing them out. We corrected them.

---

### Official Review · AnonReviewer2 · 2020-10-27
**good empirical result but with possible conceptual and computational issue**

**Rating:** 4
**Confidence:** 3

**Review:**

The current work introduces a pre-conditioning (PC) layer using low-degree polynomials as a way to implement spectral normalization (SN) of the weight matrix W in the D-net. The primary application is to improve the training of GANs by SN techniques, which has been previously established in [1] and later works like [2].

The empirical results measured by FID are good, and the experiments on higher resolution datasets like LSUN 256x256 is appreciated. However, the reviewer would like to point out some possible issues as below.

- The first issue is the conceptual relation between matrix condition number and the GAN training instability. The condition number is the ratio of the largest and smallest singular value, and thus only depends on the extreme values (the "edge") of the spectrum. Meanwhile, previous works have already shown that controlling the whole spectrum is important and beneficial for GAN training, e.g. Fig 1 in [2]. Thus it is a question whether condition number is the right object to focus on, which motivates the PC approach.
The authors show that SN-GAN can have large condition numbers, and the PC-GAN improves it (Fig. 1), which accompanies improved FID score. However, this does not mean that it is the smallest singular value, not the whole spectrum, that brings the benefit. For example, when all the singular values are 1 except for the smallest one which is 1e-8, then the matrix is almost full rank and well-conditioned except for 1 direction, but the condition number is 1e8. Suppose the spectrum of W is like this, will the training instability remain?
The theory in Section 2 is about a simplified model (linear network) under another setting, and the reviewer is not convinced that the result there (convergence is upper bounded by a quantity related to the condition number) can resolve the issue and justify the importance of condition number.

- Another possible issue is computational cost. First, the non-adaptive PC proposed is p(AA^T)A, where p is a polynomial. Even p is low-degree, this involves large matrix multiplication. Note that in the case of ill-conditioning, AA^T squares and worsen the condition number.
Second, the best empirical improvement comes from the adaptive PC, and less evident with the fixed PC. The APC method needs to compute the value of the modified condition number, and that seems to involve eigen decomposition (e.g. SVD or approximate SVD of the matrix A), which is usually expensive.
So how will the PC approach compare with the SVD like approach such as in [2] which directly controls the spectrum? It is not clear if the extra computational cost, especially eigen computation, can be justified in the practice of GAN training.

Other comments:

 Inception score in the experimental results should also be reported in the main text.

 Thm 1, Eqn (2), is it \mu/\beta or \beta/\mu?

[1] Miyato et al. "Spectral normalization for generative adversarial networks." In ICLR, 2018.

[2]  Jiang et al. "On computation and generalization of generative adversarial networks under spectrum control." In ICLR, 2019.

---

> ### Author Response · Authors · 2020-11-13
> **Response to Review #2 (continue)**
>
> **ii) Comment:** “First, $p(AA^T)A$, where p is a polynomial involves large matrix multiplication. Second, APC seems to involve eigendecomposition, which is usually expensive. ...It is not clear if the extra computational cost, especially eigen computation, can be justified in the practice of GAN training.”
>
> **Response:** We discuss the efficient implementation at the end of Sec. 3 and in Appendix C.3 and C.4. We wrote those to address the two concerns raised by the reviewer.
> (1) As Appendix C.3 explains, for computing $p(AA^T)A$, we apply three tricks: 1. computing the more efficient one of either $p(AA^T)A$ or $Ap(A^TA)$ considering the length and width of A; 2. store $AA^T$ or $A^TA$ to avoid duplicated computation; 3. use Horner’s method. With these three tricks, PC-GAN with degree 3,5,7,9 only uses 20 − 30% more time than SN-GAN. See Fig. 4a.
> (2) As Appendix C.4 explains, for APC, we only perform SVD every 1000 iterations. This contributes less than 1% of the training time. The extra running time of APC-GAN compared to SN-GAN is no more than 10%. See Fig. 4b.
> ---
> **iii) Comment:** The theory in Section 2 is about a simplified model (linear network) under another setting, and the reviewer is not convinced that the result there (convergence is upper bounded by a quantity related to the condition number) can resolve the issue and justify the importance of condition number.
>
> **Response:** Our understanding is that this concern is related to the condition number v.s. the spectrum, so we hope our response in (i) has addressed this concern.
>
> Nevertheless, we would like to add one more response to the comment “the result is just for a linear network”: We understand that expectation is perhaps subjective to individual reviewers: some may expect more from theory while some may expect more from empirical results. We also understand there is still a gap between the theory and empirical practice in our work, which is common in the optimization field. Here we would like to stress the contribution that we wish to deliver in this paper: 1)Theoretically, we build a relation between the weight spectrum and the NTK spectrum, which serves as motivation for algorithmic development; 2) Empirically, we propose the precondition layer and show its improvements on a wide variety of models, datasets and hyper-parameters. Thus, we self-position our work as a balanced combination of empirical and theoretical side.
>
> We thank the reviewer for the suggestions from the theoretical side. We believe extending to non-linear neural network is an excellent direction for future research. And we will rephrase our introduction to state the contributions con more clearly to the readers.
>
> ---
> **iv) Comment:** Inception score in the experimental results should also be reported in the main text.
>
> **Response:** Thanks for pointing this out. We have Inception Score (IS) for CIFAR-10 and STL-10 in the Appendix. We will move those to the main text in the revised version.
> IS is not good for LSUN, because IS does not work well for classes beyond ImageNet categories (e.g., LSUN), as [R1] points out in Sec 3.3 and Sec 3.6.
> ---
> **v) Comment:** Thm 1, Eqn (2), is it $\mu/\beta$ or $\beta/\mu$?
>
> **Response:** It is $\mu / \beta$. Note that $\beta > \mu$ results in $(1 - \beta/\mu)$ being negative.  Hence it can’t be correct.
>
> [R1] Qiantong Xu, Gao Huang, Yang Yuan, Chuan Guo, Yu Sun. An empirical study on evaluation metrics of generative adversarial networks. arXiv preprint

---

> ### Author Response · Authors · 2020-11-13
> **Response to Review #2**
>
> We thank the reviewer for the detailed feedback. The reviewer might have missed a few points of our paper and the preconditioning literature, which we clarify below. Thanks a lot for noting.
>
> **i) Comment:** "conceptual relation between condition number and GAN training instability"; “the matrix is almost full rank and well-conditioned except for 1 direction, but the condition number is 1e8”; “not clear whether condition number is the right object to focus on, which motivates the PC approach”
>
> **Response:** We agree that the condition number doesn’t fully determine the convergence and that the whole spectrum is the key. We were aware of it and we think our method addresses it as described next:
>
> (1) Firstly, as you also pointed out, we were aware that the traditional condition number $\sigma_{max} / \sigma_{min}$ is not a precise indicator. Thus, we use the modified condition number $\sigma_{max}  /$ (average of smallest 10% singular values) (as defined on page 6, 3rd to the last line) to decide whether we apply PC or not in the adaptive experiments. This modified condition number considers the spectrum distribution rather than just the smallest and largest singular values. Apparently, based on this metric, our APC wouldn’t apply the preconditioner in the example you raised. It is possible to design a better metric that follows Chen’05, Sec 1.5, but for practical purposes, we find our method to yield good results.
>
> (2) Secondly, in FPC, our preconditioner improves the whole spectrum, not just the condition number. For instance, if the original spectrum is [1, 0.1, 0.1, 1e-8], our preconditioner changes it to [1, 0.2, 0.2, 2e-8] which has a better spectrum, instead of modifying it to something like [1, 0.05, 1e-3, 4e-8] whose condition number is better but the spectrum is worse for convergence.
>
> Note that the gap between condition number and the spectrum is long known in numerical analysis. Johnson et al.’83 wrote: “the optimal polynomial preconditioner M (which achieves the best condition number in a certain set) may map small eigenvalues of A into large ones of $M^{-1}A$. This fact seems to degrade the convergence rate of the iteration scheme. As an alternative choice, we consider minimizing some quadratic norm....”. In other words, their viewpoint is the same as yours, which is why they propose using least-square preconditioners. We borrowed their least-square preconditioner.
>
> We understand that the word “preconditioning” may be reminiscent of the condition number. However, as a historical note, “preconditioning” is intended to improve the whole spectrum in the broad sense (since its first use by A. Turing in 1948), though the name only involves “conditioning”. In fact, the book by Chen’05 on matrix preconditioning started outlining the book by saying: “the two most relevant terms in preconditioning: condition number and clustering” (page xvi). Then the book discussed the importance of the whole spectrum (using the lens of “clustering”) in Sec 1.5. It would be more precise to change the name of the book and our method to “pre-’improving-spectrum’-er”; nevertheless, we follow the conventional terminology and use “pre-condition-er”.
>
>
> (3) That being said, there is indeed a gap between theory and practice: we prove results on condition numbers, but then improve the spectrum in practice. This gap exists in the optimization area for a long time. Most classical papers just present a result on the condition number such as “$O( \kappa \log 1/\epsilon)$ iteration complexity”, and design an algorithm to improve this bound. For instance, Johnson et al.’83 claim they design a method to improve such a bound on the conjugate gradient method, and Nesterov’s acceleration is designed to improve such a bound on GD. Their papers have a similar gap. The numerical analysis area seems to have accepted this gap, i.e., most researchers know that it is the spectrum that ultimately matters, but they often just prove results on the weaker metric “condition number”. It would be great if someone can close this gap.
>
> In summary, the common practice is: prove a theorem on the condition number ---> design a method to improve spectrum ---> refer to it as “preconditioner”. It is widely used in optimization and numerical analysis, e.g., Johnson et al.’83, and we followed this practice. The name “preconditioning” and the theorem on condition numbers might have caused the confusion, and we will add this discussion to try to clarify. Any feedback is very welcome.

---

### Official Review · AnonReviewer3 · 2020-10-28
**Comments**

**Rating:** 5
**Confidence:** 3

**Review:**

*Summary:

This paper mainly solves the instability issue on the spectral normalization for generative adversarial networks (SN-GANs) when training with high dimensional data. To address this, the authors present a preconditioning layer (PC-layer) with two different ways (i.e., FPC and APC) to perform a low-degree polynomial preconditioning. Experiments on LSUN 256x256 training data demonstrate that FPC and APC are able to control the strength of preconditioning. My detailed comments are as follows.

*Positive points:

1. The instability of training GANs is an important research problem. This paper relieves this issue by introducing a preconditioning layer.

2. This paper provides an empirical study on the training instability of GANs. The authors found that the instabilities are accompanied by large condition numbers of the discriminator weight matrices.

*Negative points:

1. Some theoretical evidence should be provided. In Section 1, the authors suspect that an ‘adequate’ weight matrix spectrum is also important for GAN training. Could you please provide sufficient theoretical analysis for this?

2. This paper mainly addresses the instability issue on SN-GANs when training with high dimensional data. However, it is not clear why SN-GANs have poor performance on higher-dimensional data. It would be better to provide theoretical analysis to support this.

3. Some technical details are not clear. The use of notations in this paper is confusing since they are used without clear explanations, which makes the paper hard to follow. For example, is $\| \|$ L2-norm by default? Should $F(\theta; x)$ and $F(\theta; X)$ be the same? Is the division in $W/ \tidle{W}$ an element-wise division? In addition, it would be better to provide intuitive understanding for the proposed theorems and claims.

4. In Section 2, this paper only considers a deep linear neural network. However, it is impractical for some real-world case. Could you please extend the analysis to more general case on deep non-linear neural network? In addition, the optimization problem mainly addresses the weighted least-square problem. Can it be extended to other losses, e.g., cross entropy loss?

5. In the experiment, the authors improve the training instability on LSUN 256x256 data. Could you please conduct more experiments on higher-dimensional dataset, e.g., CelebA 512x512 data.

---

> ### Author Response · Authors · 2020-11-24
> **Response to Review #3 (Continue)**
>
> **iii) Comment:** “Some technical details are not clear; provide intuitive understanding for the proposed theorems”
>
> **Response:** Thanks for the suggestions. In Sec 1.2 we explain that $||.||_2$ means spectral norm. Following convention, we use $||X||_F$ to denote the Frobenius norm for matrix X, and we let $||e||$ denote the standard Euclidean norm for vector e. We defined “$F(\theta, x)$” in Eq. (1), and we defined $F(\theta; X)$ in the third paragraph of Sec 2. They are different.
> We’ll add the proof sketch for Thm 1 in the revised version to help the understanding. The claims on eigenvalues are standard linear algebra results; see, e.g., [R2 Theorems 7.6 and 7.7].
>
> ---
>
> **iv) Comment:** “can you extend the analysis to nonlinear networks”; “can you extend to cross-entropy loss”.
>
> **Response:** Extending the theory to more general cases is not our focus. It often takes a number of papers in the area to gradually move forward. Even just for the optimization analysis of deep linear networks, there are more than 20 papers in the area ([Hu et al.’20] is one of them), and yet we do not fully understand deep linear networks.   Note that [Hu et al.’20, ICLR] justify orthogonal initialization using a result for a linear network.
>      That being said, we discuss a possible approach to extend to nonlinear networks. Recent breakthroughs in deep neural-nets such as Du et al.’19 and Allen-Zhu et al.’19 prove the convergence of GD to global minima for wide neural nets. Their basic lemma is that as long as the NTK matrix stays positive definite, then the loss has a sufficient reduction, leading to global convergence. This basic lemma applies to any neural net. In our work, we relate the weight matrix eigenvalues to the NTK eigenvalues in linear networks. It is possible to prove a similar relation for nonlinear networks, using techniques from Du et al.’19 and Allen-Zhu et al.’19. This may however require substantial work and it is not our focus.
>      Extending to cross-entropy loss is not easy, due to the nature of the NTK analysis. In fact, this is one of the major outstanding questions in the theoretical deep learning area. It is great that the reviewer noticed this outstanding question. Due to the difficulty of proving cross-entropy results, people choose to shrink the gap between theory and practice in a different way: Hui & Belkin’20 arXiv:2006.07322 empirically justify that quadratic loss is comparable to CE loss.
>
> ---
>
> **v) Comment:** “more experiments on CelebA 512x512”
>
> **Response:** Thanks for the suggestion. We run the experiments on the CelebA 512x512 with CNN. We use a similar structure as Tab.9 stated in the Appendix, and keep all other hyper-parameters the same as before. The results at the 20k iteration are:
>
> Model             $\qquad$          FID(smaller is better)
> SN-GAN            $\hspace{1.8cm }$             29.44
> SVD                    $\hspace{2.4cm }$                 31.68
> PC-GAN-deg3     $\hspace{1cm } $                21.39
> APC-GAN             $\hspace{1.65cm }$           **20.91**
>
> where PC-GAN and APC-GAN outperform SN-GAN and SVD.
>
>
> [R1] Towards a Better Global Loss Landscape of GANs, Sun, R., Fang, T., & Schwing, A. (2020). Towards a Better Global Loss Landscape of GANs. Advances in Neural Information Processing Systems, 2020.  https://arxiv.org/pdf/2011.04926.pdf
>
> [R2] Matrix Theory and Applications for Scientists and Engineers. Courier Dover Publications. Graham, A., 2018.

---

> ### Author Response · Authors · 2020-11-24
> **Response to Review #3**
>
> Thank you for your helpful comments.
>
> We understand that the expectation of a paper is perhaps subjective to individual reviewers: some may expect more theory while some may expect more empirical results. Here we would like to stress that our contribution is theory-inspired algorithmic development: the theory mostly serves as motivation for algorithmic development. Thus, we position our work as a combination of algorithm and theory. We thank the reviewer for the suggestions from the theoretical side, and we believe they are excellent directions for future research.
>
> We understand there is still a gap between theory and practice in our work. However, it is a common phenomenon in the deep learning optimization field, and we believe our work has a smaller gap than most previous works. Specifically, We suggest a “good weight spectrum improves training”, which extends Hu et al.’20 who show that the initial weight spectrum is important for training: we prove that controlling the spectrum throughout training helps convergence. Our work and Hu et al.'20 are extensions of earlier understanding "spectrum of Jacobian is important for training" and "spectrum of the NTK is important for training".
>
> ---
>
> Please see below for our response to your specific concerns.
>
> **i) Comment:** Theory for “‘adequate’ weight matrix spectrum is also important for GAN training”:
>
> **Response:** Our understanding of the comment is the following: the current paper proves a result of supervised learning while the simulation is for GAN, which exhibits a gap. We agree that there is a gap. Nevertheless, our intention is that PC-layer can improve any neural network training, thus can likely improve GAN training. Our PC-layer is like BatchNorm: the original paper of BatchNorm is designed for general use with simulation on image classification, but nonetheless, it is immediately applicable to GANs and other tasks.
>
> Analyzing the convergence of GAN is an interesting question. There is little work in the area that analyzes the convergence of GAN, since the nonconvex min-max problem is notoriously difficult to analyze. Recent works on GAN convergence are often for very special settings, e.g. linear networks for learning a single-point distribution https://arxiv.org/pdf/1801.04406.pdf (ICML'18) and single-layer-single-neuron-network for learning a linear-model-generated-distribution https://arxiv.org/abs/1910.07030 (ICML'20). A result for GAN that involves just a 2-layer neural net and yet relevant for practical training is already challenging.
>
> That being said, below we point out a possible approach based on a very recent work [R1] that will appear in NeurIPS'2020. It proposed a framework for analyzing GANs that consists of two components: an analysis of the GAN loss in the function space, and an analysis that relates the neural-net parameter space and the function space. A possible way to combine their framework and ours is the following: first, a well-conditioned weight spectrum implies a well-conditioned NTK; second, as their Claim K.1 and Claim K.2 state, a full-rank NTK implies that Assumption 4.6 (which connects neural-net parameter space to function space) holds; third, as their Appendix K proves, Assumption 4.6 implies that a result in neural net parameter space holds. A formal result and formal proof probably require another paper to describe, and we will leave it to future work.
>
> ---
>
> **ii) Comment:** it is not clear why SN-GANs have poor performance on higher-dimensional data; better to provide a theoretical analysis
>
> **Response:** As we mentioned in the paper, SN cannot avoid singular weight matrices, and as a result cannot avoid a singular NTK matrix. The foundational lemma of NTK theory (also see our Lemma 1) states that a non-singular NTK ensures global convergence, thus implying that a local minimum with singular NTK may be a bad local minimum. Thus it is not surprising that SN can fail for some difficult cases.
> However, why does SN succeed for low-dimensional data? Intuitively, low-dimensional problems are ‘easy,’ so the risk of running into a singular NTK is smaller. Lin et al.’20 (pointed out by Reviewer 5) show that SN can avoid gradient explosion. Note that no-gradient-explosion is weaker than a well-conditioned NTK. Thus SN only ensures a weaker property (no gradient-explosion), and a well-conditioned spectrum ensures a stronger property (local-min is global). Thus SN works for some cases like low-dimensional data, and PC works for more cases including high-dimensional data.
>    In short, "singular NTK points can be bad local minima" + "SN cannot avoid singular NTK" ==> SN may lead to bad local minima. We hope this explanation can shed some light on why SN can fail. Note, however, that the risk of failure in high-dimensional image generation exists for most existing tricks, including SN, BN or other normalization methods. They are not designed to ensure full-rank NTK, thus it is possible that they fail.

---

### Official Review · AnonReviewer4 · 2020-10-28
**New idea, no demonstration of improved visual quality**

**Rating:** 6
**Confidence:** 3

**Review:**

The paper proposes a new concept of pre-conditioning layers. The authors claim that this idea is novel and they introduce this idea in the context of GANs. I cannot verify or falsify if the idea is really novel, but assuming the idea is novel, I definitely believe it is worthy to be explored and eventually published somewhere. The idea is creative and technically interesting.

The authors argue that not only large singular values matter, but also the condition number.
Based on their experiments, the author find that a bad condition number might be a reason for poor results using spectral normalization in a GAN.

The technical realization of the idea seems reasonable.

The evaluation is a bit difficult to follow. The baseline method is a GAN that is no-longer state of the art. It is not even clear to me what this GAN architecture is exactly that is being used. The description is a bit vague. I am not sure if the baseline architecture can be considered reasonable. Certainly, while spectral normalization might cause certain problem with one type of architecture, it is not that clear that this problem will persist in a state-of-the-art GAN and how many state-of-the-art GANs really rely on spectral normalization. For example, the StyleGANv2 paper writes
"It should be noted that spectral normalization [31] of the generator [46] only constrains the largest singular value, posing no constraints on the others and hence not necessarily leading to better conditioning. We find that enabling spectral normalization in addition to our contributions — or instead of them — invariably compromises FID, as detailed in Appendix E"

Further, just relying on FID is risky. If there is no demonstration of visual quality improvement I would be quite skeptical if an improvement really has been made. It is ultimately up to the authors to demonstrate an improvement. With this submission, the authors shift a massive amount of experimental validation of the work to the reader. Who is going to try out your idea on a state-of-the-art GAN and evaluate visual quality?

---

> ### Author Response · Authors · 2020-11-24
> **Response to Review#4 (Continue)**
>
> **ii) Comment:** StyleGAN2 mentions that “It should be noted that spectral normalization [31] of the generator [46] only constrains the largest singular value, posing no constraints on the others and hence not necessarily leading to better conditioning”.
>
> **Response:** We agree that SN only constrains the largest singular value and not necessarily improves the conditioning. That is exactly our motivation. This comment of StyleGANv2 is consistent with our message that SN could be further improved, which is why we propose the PC-layer.
>
> ---
>
> **iii) Comment:** Further, just relying on FID is risky. If there is no demonstration of visual quality improvement I would be quite skeptical if an improvement really has been made.
>
> **Response:** Thanks for the comment.
>
> First, we include the generated images at the end of the Appendix. Particularly, compared to SN-GAN we show qualitative improvements of PC-GAN and APC-GAN on CIFAR-10 (Fig.8 1st column) and STL-10 (Fig.10 1st column) with D_iter=1 and LSUN-bedroom (Fig. 11) and LSUN-living-room (Fig. 12). We also reported IS scores for CIFAR-10 and STL-10 in Appendix Tab. 4.
>
> Second, to further address this concern, we add MMD as an extra metric; as a complementary for Tab. 1 and Tab. 2 (LSUN-bedroom 256):
>
> CIAFR-10 on ResNet witn D_{iter}=1
> $\hspace{2cm} $   FID (smaller is better)    $\hspace{0.5cm}$      MMD (smaller is better)
> SN-GAN      $\hspace{2cm}$               77.82            $\hspace{2.5cm} $                    0.122
> SVD                   $\hspace{2.6cm}$           20.75              $\hspace{2.5cm} $                   0.049
> FPC, deg-3        $\hspace{1.6cm}$         20.09            $\hspace{2.5cm} $                    0.051
> FPC, deg-7    $\hspace{1.6cm}$         **19.31**          $\hspace{2.4cm} $                  0.050
> APC             $\hspace{2.6cm}$               19.53           $\hspace{2.5cm} $                 **0.045**
>
>
> STL-10  on ResNet witn D_{iter}=1
> $\hspace{2cm} $   FID (smaller is better)    $\hspace{0.5cm}$      MMD (smaller is better)
> SN-GAN      $\hspace{2cm}$             147.90           $\hspace{2.4cm} $                 0.178
> SVD            $\hspace{2.6cm}$              38.01            $\hspace{2.5cm} $                   0.0782
> FPC, deg-3      $\hspace{1.6cm}$           33.99         $\hspace{2.5cm} $                      0.0725
> FPC, deg-7        $\hspace{1.6cm}$         34.28          $\hspace{2.5cm} $                     0.0820
> APC            $\hspace{2.6cm}$           **34.08**       $\hspace{2.4cm} $                **0.0669**
>
> LSUN bedroom 256
> $\hspace{2cm} $   FID (smaller is better)    $\hspace{0.5cm}$      MMD (smaller is better)
> SN-GAN       $\hspace{2cm}$            81.09             $\hspace{2.5cm} $               0.153
> SVD         $\hspace{2.6cm}$                  36.89          $\hspace{2.5cm} $                  0.105
> FPC, deg-3      $\hspace{1.6cm}$          35.61        $\hspace{2.5cm} $                    0.106
> APC         $\hspace{2.6cm}$             **31.17**      $\hspace{2.4cm} $               **0.103**
>
> These results show that our APC and FPC still significantly improve upon SN-GAN using a different metric MMD.

---

> ### Author Response · Authors · 2020-11-24
> **Response to Review#4**
>
> Thank you for your comments. Please see the response below.
>
> **i) Comment:** The evaluation is a bit difficult to follow. The baseline method is a GAN that is no-longer state of the art. It is not even clear to me what this GAN architecture is exactly that is being used. StyleGANv2 paper writes “spectral normalization [31] of the generator [46] … invariably comprises FID“. How many state-of-the-art GANs really rely on spectral normalization?
>
> **Response:** We believe using SN (within the discriminator) is a good choice for the following reasons.
> (1) Investigating StyleGAN2 results, SN on the discriminator (SN-D) slightly degrades FID, but improves PPL. More specifically, according to the 1st and 2nd row of Table 4 in the StyleGANv2 paper, adding SN-D slightly degrades FID (2.83 to 2.98), but improves PPL (145.0 to 131.4). Further adding SN-G (SN on generator) would lead to an even worse FID (3.40) but slightly better PPL (131.4 to 130.9). If we want to achieve a good trade-off between FID and PPL, the table suggests that using SN-D but not SN-G is the best choice: adding SN-G causes a 0.42 FID degradation, while removing SN-D causes 15 PPL degradation.
> In short, just using SN-D but not SN-G, as our paper and SN-GAN did, is a good choice even by the StyleGAN2 experiment results.
>
> (2) Spectral Normalization is the technique widely studied in many recent GAN works. We find that out of 24 GAN related papers in ICLR 2020, there are 12 papers that either apply SN in their proposed model or study SN. We list those papers below:
> * Casey Chu, Kentaro Minami, Kenji Fukumizu. Smoothness And Stability In Gans
> * Michel Besserve, Arash Mehrjou, Rémy Sun, Bernhard Schölkopf. Counterfactuals Uncover The Modular Structure Of Deep Generative Models
> * Ali Jahanian, Lucy Chai, Phillip Isola. On The “Steerability” Of Generative Adversarial Networks
> * Mingrui Liu, Youssef Mroueh, Jerret Ross, Wei Zhang, Xiaodong Cui, Payel Das, Tianbao Yang. Towards Better Understanding Of Adaptive Gradient Algorithms In Generative Adversarial Nets
> * Hugo Berard, Gauthier Gidel, Amjad Almahairi, Pascal Vincent, Simon Lacoste-Julien. A Closer Look At The Optimization Landscapes Of Generative Adversarial Networks
> * Antoine Plumerault, Hervé Le Borgne, Céline Hudelot. Controlling Generative Models With Continuous Factors Of Variations
> * Ari Holtzman, Jan Buys, Li Du, Maxwell Forbes, Yejin Choi. The Curious Case Of Neural Text Degeneration
> * Junho Kim, Minjae Kim, Hyeonwoo Kang, Kwanghee Lee. U-gat-it: Unsupervised Generative Attentional Networks With Adaptive Layer Instance Normalization For Image-to-image Translation
> * Daniel Stoller, Sebastian Ewert, Simon Dixon. Training Generative Adversarial Networks From Incomplete Observations Using Factorised Discriminators
> * Han Zhang, Zizhao Zhang, Augustus Odena, Honglak Lee. Consistency Regularization For Generative Adversarial Networks
> * Yuanbo Xiangli, Yubin Deng, Bo Dai, Chen Change Loy, Dahua Lin. Real Or Not Real, That Is The Question
> * Amartya Sanyal, Philip H.S. Torr, Puneet K. Dokania. Stable Rank Normalization For Improved Generalization In Neural Networks And Gans
>
> Besides, SN is used in BigGAN which achieves the SoTA result on the ImageNet 128x128 task.
>
> (3) We do not use StyleGAN and BigGAN due to the large computational cost. BigGAN requires two weeks of training with 8 V100 on ImageNet 128x128, and StyleGANv2 requires about 32days to train on 256*256 FFHQ dataset on one V100. We don’t have those computational resources: each instance we reported requires about 1-1.5 days of training on a single 2080Ti GPU. If we use a 2080Ti GPU to train BigGAN or StyleGAN, it would take about 30 days.
>
> (4) We suspect BigGAN, which uses SN on both the generator and the discriminator also suffers from the problem of a bad spectrum distribution. In their Fig. 3, they observe the largest singular value $\sigma_{max}$ explode for both G and D. Our method could provide a possible solution (though due to limited compute, we could not check PC on BigGAN).

---

### Decision · Program_Chairs · 2021-01-07
**Final Decision**

**Decision:**

Reject

**Comment:**

This paper presents a new method for training GAN by adding a precondition Layer. All reviewers are positive about the empirical results. However, some concerns were raised about the justification: (1) Only linear networks are considered, which is a bit impractical; (2) Existing work has shown the importance of controlling the whole spectrum instead of the condition number. There should be some connection missing between the proposed result and existing results; (3) The computational cost is a bit high. The paper would be much stronger if these concerns could be addressed.

---

> ### Author Response · Authors · 2021-01-31
> **Response to Program Chair**
>
> We thank the PC for feedback, and we appreciate the time and effort in handling our paper. For future readers, we would like to amend points (2) and (3): (2) Note, analogous to our work, the preconditioning literature is motivated by the fact that an improved condition number leads to a better convergence rate in theory, while their designed preconditioners improve the spectrum in practice. (3) We believe that our computation is efficient: as we show in the Appendix, we only add an extra 10% time upon SN-GAN, which includes the time for calculating the condition number.